# Automation and control of laser wakefield accelerators using Bayesian optimization

R. J. Shalloo [1✉], S. J. D. Dann [2], J.-N. Gruse [1], C. I. D. Underwood [3], A. F. Antoine [4], C. Arran[3],
M. Backhouse[1], C. D. Baird[2,3], M. D. Balcazar[4], N. Bourgeois[2], J. A. Cardarelli[4], P. Hatfield [5], J. Kang[6],
K. Krushelnick[4], S. P. D. Mangles [1], C. D. Murphy [3], N. Lu[7], J. Osterhoff [8], K. Põder [8], P. P. Rajeev [2],
C. P. Ridgers[3], S. Rozario[1], M. P. Selwood [3], A. J. Shahani[7], D. R. Symes[2], A. G. R. Thomas [4], C. Thornton [2],
Z. Najmudin [1] & M. J. V. Streeter [1]

Laser wakefield accelerators promise to revolutionize many areas of accelerator science. However, one of the greatest challenges to their widespread adoption is the difficulty in control and optimization of the accelerator outputs due to coupling between input parameters and the dynamic evolution of the accelerating structure. Here, we use machine learning techniques to automate a 100 MeV-scale accelerator, which optimized its outputs by simultaneously varying up to six parameters including the spectral and spatial phase of the laser and the plasma density and length. Most notably, the model built by the algorithm enabled optimization of the laser evolution that might otherwise have been missed in single-variable scans. Subtle tuning of the laser pulse shape caused an 80% increase in electron beam charge, despite the pulse length changing by just 1%.

[1] The John Adams Institute for Accelerator Science, Imperial College London, London SW7 2AZ, UK. [2] Central Laser Facility, STFC Rutherford Appleton Laboratory, Didcot OX11 0QX, UK. [3] Department of Physics, York Plasma Institute, University of York, York YO10 5DD, UK. [4] Center for Ultrafast Optical Science, University of Michigan, Ann Arbor, MI 48109-2099, USA. [5] Clarendon Laboratory, University of Oxford, Parks Road, Oxford OX1 3PU, UK. [6] Department of Chemical Engineering, University of Michigan, Ann Arbor, MI 48109, USA. [7] Department of Materials Science and Engineering, University of Michigan, Ann Arbor, MI 48109, USA. [8] Deutsches Elektronen-Synchrotron DESY, Notkestraße 85, 22607 Hamburg, Germany. ✉email: r.shalloo@imperial.ac.uk

In a laser wakefield accelerator (LWFA), an ultrashort intense laser pulse travelling through a plasma creates a wave in its wake, which can be used to accelerate electrons to multi-GeV energies in just a few centimetres[1]. The enormous accelerating fields achievable in LWFAs could dramatically reduce the size and cost of future high-energy accelerators[2]. In addition, the X-rays generated by transverse oscillations of electrons trapped within the plasma structure can provide compact ultrafast synchrotron sources[3,4]. As such, there are a number of facilities based on LWFAs at various stages of planning, construction and operation[5–7]. In addition, there is now a global effort aimed at designing a compact plasma-based particle collider in lieu of, or even superseding, a future multi-10 km-scale linear accelerator based on conventional technology[8].

In an LWFA, the laser pulse drives the plasma wave via the ponderomotive force, which depends on laser intensity, shape and spectral content. In general, all of these parameters are constantly evolving throughout the acceleration process. This is particularly evident in strongly driven LWFAs where electrons are accelerated from within the plasma itself[9,10]. Although it is possible to obtain simple expressions for the dependence of LWFA output on plasma density and laser intensity for an unchanging laser pulse[11], in reality, the evolution of laser parameters makes analytical treatment less tractable.

Furthermore, there are a large number of input parameters that must be tuned to optimize the accelerator performance, including those which affect the spatial and spectral energy distribution of the laser pulse and those which control the nature of the plasma source. The usual approach to optimization is to perform a series of single variable (one-dimensional, 1D) scans in the neighbourhood of the expected optimal settings. These scans are challenging, as the input parameters are often coupled and the highly sensitive response of the system can lead to large shot-to-shot variations in outputs. Moreover, due to the non-linear evolution of the LWFA, altering one input can affect the optimal values of all the other input parameters. Hence, sequential 1D optimizations do not reach the true optimum unless initiated there. A full $N$-D scan would be prohibitively time consuming for $N > 2$ and so a more intelligent search procedure is required.

Machine learning techniques are ideal for this kind of problem and have been demonstrated in other plasma physics, accelerator science and light source applications[12–15]. Genetic algorithms

have been applied to laser-plasma sources including; using the spatial phase of the laser to optimize a keV electron source[16], and subsequently using both spectral and spatial phase (although not simultaneously) to optimize a MeV-electron source[17]. In both cases, only the laser parameters were controlled preventing full optimization of the LWFA which relies on the complex interplay between the laser and the plasma. Further, these optimizations did not incorporate the experimental errors and were therefore prone to distortion by statistical outliers.

In this study, we present the use of Bayesian optimization to demonstrate operation of the first fully automated laser-plasma accelerator. Simultaneous control of up to six laser and plasma parameters enabled independent optimization of different properties of the source far exceeding that achieved manually with a 5 TW class laser system[18]. In performing the optimization, the algorithm builds a surrogate model of the parameter space, including the uncertainty arising from the sparsity of the data and the measurement variances.

## Results

**Bayesian optimization.** Bayesian optimization is a popular and efficient machine learning technique for the multivariate optimization of expensive to evaluate or noisy functions[19,20]. At its core, it creates a surrogate model of the objective function (the observable to be optimized), which is then used to guide the optimization process. The model is a prior probability distribution over all possible objective functions, representing our belief about the function's properties such as amplitude and smoothness. This distribution is commonly realized as a Gaussian process model in a technique called Gaussian Process Regression (GPR)[21]. The prior distribution is updated with each new measurement to produce a more accurate posterior distribution. The mean of this distribution (the black line in Fig. 1b–e) is our best estimate of the objective function's form (the red dashed line in Fig. 1b–e) and its maximum gives the best estimate of the maximum of the real objective function.

Every function sampled from the posterior distribution will be compatible with the measurements used to construct it. In the case where the measurements have some variance associated with them, this information can also be fed into the posterior distribution, such that functions sampled from the posterior

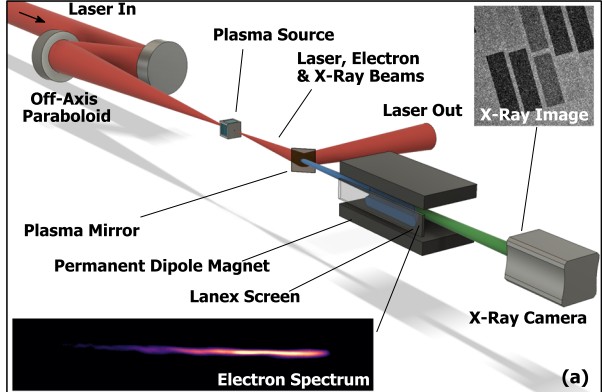

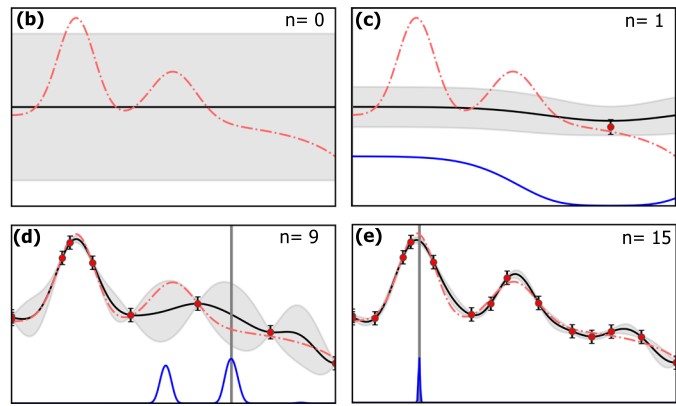

**Fig. 1 Experimental schematic and principle of Bayesian optimization algorithm. a** Schematic of the laser-plasma accelerator showing an ultrashort, intense laser pulse focused into a plasma source. The laser drives a relativistic plasma wave, accelerating electrons to MeV energies and producing keV X-rays. The spectral and spatial phase of the laser pulse prior to focusing, and density and length of the plasma source could all be controlled programatically. The electrons and X-rays were diagnosed by a permanent-dipole magnet spectrometer and a direct-detection X-ray camera, respectively. **b–e** 1D example of Bayesian optimization algorithm with $n = 0$, 1, 9, and 15 measured data points. The dashed red line represents the true function, the black line represents the surrogate model and the grey shaded region denotes the standard deviation of the model. Data points are sampled from the true function with some simulated measurement error (red circles). As more data points are added, the surrogate model begins to more closely resemble the true function. The blue line represents the acquisition function, the maximum of which tells the algorithm which point to sample next.

distribution need only fit the data to the precision dictated by their uncertainty. Using this approach, the model includes both uncertainty and correlations between measurements at different points. The correlation between any two points in the space is characterized by the covariance kernel.

Our Bayesian optimization procedure is conceptually depicted in Fig. 1b–e and proceeded as follows:

1. A Gaussian process model is constructed, using a physically sensible form for the covariance kernel.
2. A number of experimental measurements are made at chosen positions to initialize the algorithm.
3. The model is updated with the accumulated measurements to form a posterior distribution.
4. An acquisition function is computed (see below) and used to select the next measurement location.
5. Steps 3–4 are repeated until the convergence criteria are met.

The next point to be sampled is determined by an acquisition function based on the mean $\mu$ and standard deviation (SD) $\sigma$ of the model. This allows for a trade-off between exploring parts of the domain where few measurements have been made ($\sigma$ is high) and exploiting parts of the domain believed to be near a maximum ($\mu$ is high). A simple example acquisition function is the upper confidence bound, $UCB = \mu + \kappa\sigma$, where $\kappa$ characterizes the trade-off between exploration and exploitation.

In the work performed here, an augmented Bayesian optimization procedure, developed by the authors but based on the *scikit-learn* platform[22], was utilized. This algorithm included two GPR models, to allow for efficient sampling of the parameter space in the presence of input-dependent measurement uncertainty (see 'Methods' for details).

**Experimental setup**. The experiments were performed with the Gemini TA2 Ti:sapphire laser system at the Central Laser Facility, using the arrangement shown in Fig. 1a. On target, each laser pulse contained ~245 mJ, had a 45 fs transform limit and was focused to a $1/e^2$ spot radius of 16 µm for a peak normalized vector potential of $a_0 = 0.55$. Despite its relatively modest specifications—most notably a peak power of just 5.4 TW—the laser can be used to drive a 100 MeV-class LWFA at 1 Hz, with a gas cell acting as the plasma source.

The relevant outputs of the source were measured at the exit of the plasma by standard diagnostics; an electron spectrometer to measure energy distribution, charge and beam profile of the accelerated electrons, and an X-ray camera that measures yield, energy and divergence of generated betatron X-rays.

The optimization algorithm was used to control this accelerator by manipulating the spectral and spatial phase profiles of the laser pulse as well as the length and electron density of the plasma. The spectral phase of the laser pulse $\phi(\omega)$ was controlled by an acousto-optic programmable dispersive filter, allowing for variation of the temporal profile of the compressed laser pulse. The changes to the spectral phase were parameterized by the coefficients of a polynomial $\phi(\omega) = \sum (\omega - \omega_0)^n \beta^{(n)}/n!$, with $\beta^{(n)} = 0$ corresponding to optimal compression. The second, third and fourth orders ($\beta^{(2)}$, $\beta^{(3)}$, $\beta^{(4)}$) were independently controlled by the algorithm. A piezoelectric deformable mirror provided control over the spatial phase of the laser pulse, allowing the algorithm to apply deformations to the wavefront, for example to shift the focal plane relative to the electron density profile. The electron density of the plasma was controlled by changing the pressure of a gas reservoir connected directly to the plasma source. Modification of the length of the plasma source was achieved by changing the length of the custom-designed gas cell.

Every element of the optimization, the control, analysis and selection of the next evaluation point, proceeded automatically without input from the user. For each measurement, a single burst comprising ten shots was taken. Each diagnostic was analysed and the results were used to calculate the objective function. Taking ten shots allowed for calculation of the mean and variance of the objective function for a given set of input parameters. During the optimization runs, all selected parameters were free to vary simultaneously and so were all optimized concurrently.

**Optimization of electron and X-ray production**. We demonstrate the optimization algorithm by using a simple objective function; the total counts recorded by the electron spectrometer. This corresponds to the total charge in the laser-generated electron beam with $E = \gamma m_e c^2 > 26$ MeV. To demonstrate the reliability of the optimization, 10 consecutive optimization runs were performed using the same algorithm. A gas mixture of 1% nitrogen and 99% helium was used to allow for ionization injection[23–25] providing a reduced threshold for electron beam generation compared to pure helium. The optimization varied four input parameters; the spectral phase coefficients $\beta^{(2)}$, $\beta^{(3)}$ and $\beta^{(4)}$, and the longitudinal position of focus of the laser pulse $f$. The first measurement point for each run was taken at the optimum position from the previous days operation. Due to the drift of laser performance and experimental parameters, optimal positions varied day to day.

To track the progress of the algorithm during each optimization, we obtained the surrogate model's prediction of the global maximum after each burst. The average and standard deviation of this predicted optimum over the ten runs is plotted in Fig. 2. The algorithm was able to optimize electron beam charge in 4D with just 20 measurements, consisting of 200 total shots and taking 6.5 min including the time for parameter setting and computation. In each case, the final optimum value (indicated by the dashed line) was reached after ~20 bursts, resulting in a 3 times increase in electron beam charge compared to the unoptimized starting position. After this point, the local maximum has been fully exploited and the algorithm continued to explore other parts of the parameter space where statistical uncertainty of the model was largest. The mean and SD optimized electron charge from the ten runs was $17 \pm 2$ pC.

For a more challenging optimization, we chose the yield of betatron X-rays as the objective function. Maximizing the flux of these ultrashort bursts of X-ray radiation would be of great benefit for a diverse range of applications, such as the imaging of medical, industrial and scientific samples[4]. The X-rays from an LWFA can be emitted at any point in the accelerator where the

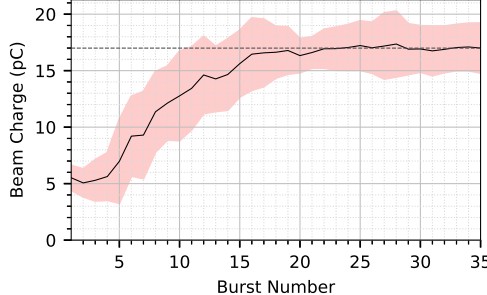

**Fig. 2 Convergence test of ten identical optimization runs.** The maximum predicted electron beam charge (for $\gamma m_e c^2 > 26$ MeV) across the ten runs is plotted as a function of burst number (ten shots per burst). The shaded region encloses a ±SD of uncertainty. The runs were performed in a nitrogen–helium gas mix.

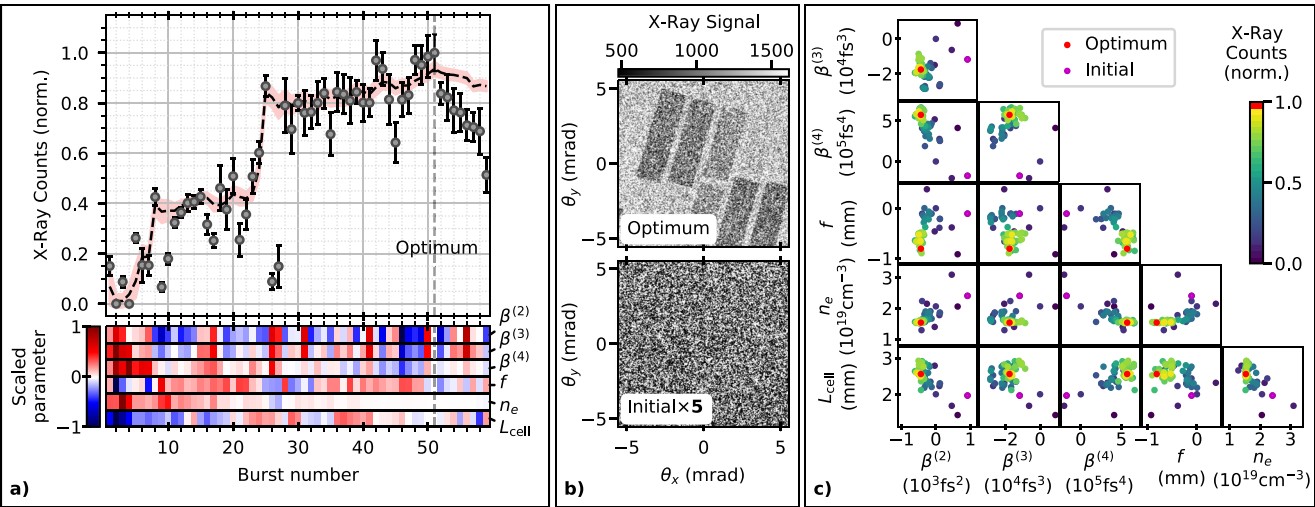

**Fig. 3 6D optimization of betatron X-ray source. a** Top panel shows the mean (normalized) X-ray yield for each burst (circles) with SE. The dashed line shows the model predicted maximum value of X-ray yield with the shaded region enclosing ±SD uncertainty. The lower panel shows the evolution of the input parameters representing the focal position of the laser pulse ($f$), the plasma electron density ($n_e$), the plasma source length ($L_{cell}$) and three orders of spectral phase ($\beta^{(2)}, \beta^{(3)}, \beta^{(4)}$). **b** X-ray images from the initial and optimal bursts, where the signal for each pixel is the mean from the ten individual shots. The initial X-ray image is multiplied by 5, to make it visible on the same scale. **c** The projection of the measurements onto each 2D plane of the parameter space, colour coded by the normalized total X-ray counts.

electrons reach a high energy and oscillate with a large amplitude. These electrons may subsequently decelerate such that they are not detected by the electron spectrometer, and so the X-ray flux may be optimized by substantially different input parameters than those that optimized the measured electron beam charge.

An example is shown in Fig. 3a, where the total X-ray yield, characterized as counts on the X-ray camera, was optimized in a pure helium plasma. Six input parameters were varied, incorporating the backing pressure and length of the gas cell. Here, a fivefold increase in X-ray yield is achieved in a 27 min 6D optimization. This results in a dramatic increase in the usability of the X-ray source, as shown in Fig. 3b, where the filter array becomes clearly visible. This is notable, as the energy of this laser system would usually be considered inadequate for betatron imaging applications in the multi-keV energy range[4].

The bottom panel of Fig. 3a shows how the input parameters were varied for each burst to achieve the indicated X-ray yield. For the purposes of this visualization, the input parameters are offset so that the optimum position is at 0 and scaled so that all values lie in the range ±1. The pair-plots of the measurement positions, shown in Fig. 3c, show how each parameter varied and were guided towards the local optimum. The initial position was the optimum from the previous days operation, for which the laser performance was significantly different, including 7% lower average pulse energy. The optimization was able to tune the laser compression and focusing, and also found increased performance by operating at a lower plasma density and longer gas cell length.

**Tailoring electron beam characteristics.** A strength of a fully automated LWFA is that the highly flexible accelerator can be tailored to specific applications merely by changing the objective function. For example, for the generation of positron beams[26] or γ-rays[27,28], it is advantageous to prioritize the conversion of laser energy to electron beam energy. By contrast, for sending the output of the LWFA to a second acceleration stage[29], fine control of the electron beam divergence and energy spread is more important. Careful selection of this objective function is vital and can be used to control the phase space of the beams.

In defining the objective function, any combination of measurable quantities may be used as long as they can be

expressed as a single number with a good estimate of the measurement error. Here, the results of two additional optimizations based on more complex objective functions are presented; one targeting the total electron beam energy (example A) and the other the electron beam divergence (example B). In both cases, the gas cell was filled with helium doped with 1% nitrogen to allow for ionization injection. The gas cell length was fixed in each example prior to optimization reducing the automated optimizations to 5D.

For example A, the initial conditions were seen to be far from optimal and during the 20 min 5D optimization, all five input parameters had to vary significantly to achieve the optimum, an average total beam energy of 0.91 ± 0.15 mJ. For example B, an objective function was employed which only summed charge within a 3.75 mrad acceptance angle around the laser axis. This rewarded electron beams with a high charge per unit divergence, which were well aligned to this central axis. This optimization gave a minimum burst-averaged electron beam divergence of 3.4 ± 0.2 mrad, whereas the total beam energy was lower than in example A at 0.26 ± 0.04 mJ.

Both optimizations achieved the maximum values of their objective functions within 40 bursts. Figure 4 shows ten consecutive beams from the best burst of each of the two optimizations. There is a clear distinction in the form of the optimal electron beams for the two cases. This demonstrates the fundamental impact that the choice of objective function has on the accelerator performance. It also shows the importance of choosing the correct objective function for a given application, as although the total beam charge for example B is lower, it is far more suitable for some applications, e.g., if the beam is required to pass through a narrow collimator to an interaction chamber.

Although the qualitative features of the beams in Fig. 4 are consistent in each of the two bursts, it is clear that there is also shot-to-shot variation in the spectra of the beams for nominally identical conditions. This variation can be primarily attributed to the stability of the laser system, which had peak-to-valley fluctuations in the pulse energy of 8%, the pulse duration of 6% and focal position of a Rayleigh length (1 mm). It is a testament to the Bayesian optimization-based approach that optima could be reliably located despite the shot-to-shot fluctuations in

parameters. Implementing these automated optimization techniques on next-generation laser systems, which demonstrate significantly higher stability in the laser parameters[30], will result in much finer tuning and control of the electron and X-ray beams.

**Exploring the models.** The model constructed by the optimization algorithm describes the behaviour of the physical system with increasing accuracy as more measurements are taken. In the case of the optimization convergence runs discussed above, 350 measurements, consisting of 3500 shots, were combined from ten runs to generate a model of the 4D parameter space. The optimal parameters of this model were as follows: 60 fs$^2$, $9 \times 10^3$ fs$^3$, $6 \times 10^5$ fs$^4$ and 0.7 mm for $\beta^{(2)}$, $\beta^{(3)}$, $\beta^{(4)}$ and $f$, respectively, relative to the starting position. By investigating this model, a clear correlation was observed between two of the input parameters, the second $\beta^{(2)}$ and fourth $\beta^{(4)}$ order components of the spectral

phase. This can be clearly seen by taking a 2D slice through the 4D parameter space at the optimal values of $\beta^{(3)}$ and $f$ as shown in Fig. 5a.

The correlation between $\beta^{(2)}$ and $\beta^{(4)}$ is a consequence of expressing the spectral phase as a polynomial, i.e., even orders are mathematically coupled. The chirp of the laser pulse due to the introduction of $+\beta^{(2)}$ can be partially compensated by $-\beta^{(4)}$, maintaining a high peak power. The change in group delay at $\pm\Delta\omega$ due to a change of $\Delta\beta^{(2)}$ is cancelled out for $\Delta\beta^{(4)} = -6\Delta\beta^{(2)}/(\Delta\omega)^2$. The solid and dashed lines in Fig. 5a represent this relationship using the measured full width at half maximum (FWHM) bandwidth and match the observed gradient of the correlation. The dashed line shows this relationship centered on $(\beta^{(2)}, \beta^{(4)}) = (0, 0)$ relative to the fully compressed pulse. The solid line passes through $(\beta^{(2)}, \beta^{(4)}) = (390 \text{ fs}^2, 0)$, representing a pulse with a small amount of positive chirp.

Along the solid line, which passes through the optimum of the trained surrogate model, the charge produced by the LWFA remains high for a significant change in spectral phase coefficients. Previous observations have also determined that a laser pulse with a small positive chirp and a steep rising edge is advantageous for self-injection in a 1D scan of $\beta^{(2)}$[31]. Here we find that a small amount of positive chirp and a steep rising edge is optimal for ionization injection also, and that subtle changes to the laser pulse shape, using a combination of $\beta^{(2)}$ and $\beta^{(4)}$, maximize this enhancement. In moving from the unchirped pulse (position **A** in Fig. 5a) to the optimal positively chirped pulse (position **B**), keeping focus and $\beta^{(3)}$ at their optimal values, we observe an 80% increase in charge. The large change in charge is remarkable, considering that the standard measure of laser pulse length, the FWHM, changed by only 0.5 fs in this optimization.

Simulations were performed using the quasi-three-dimensional particle-in-cell (PIC) code FBPIC[32] to understand the reason for the observed behaviour. The laser pulse was initialized to match experimental measurements of the transverse intensity profile and the temporal intensity and phase. The peak vacuum $a_0$ was set to 0.55, ensuring that the integral of the laser energy distribution was equal to the average pulse energy of this run (0.245 J).

After entering the plasma, the driving laser pulse undergoes self-focusing, self-modulation and self-compression (as shown in Fig. 5b) increasing the intensity of the pulse. This causes further

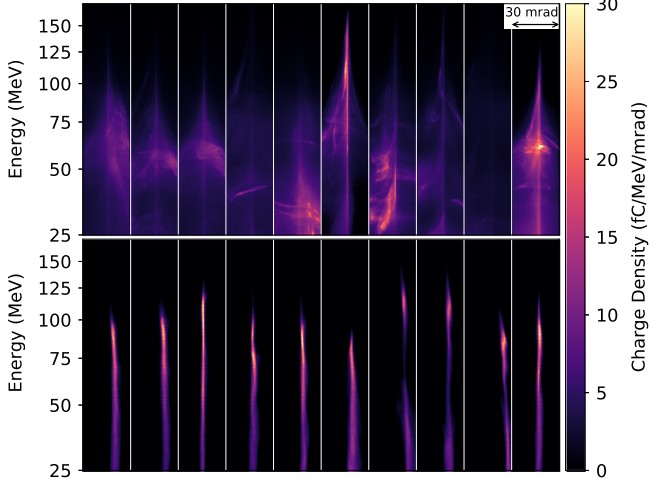

**Fig. 4 Angularly resolved electron spectra for different objective functions.** A burst (ten consecutive shots) optimized for: (example A, top) total electron beam energy; (example B, bottom) electron beam divergence obtained by counting charge within a 3.75 mrad acceptance angle.

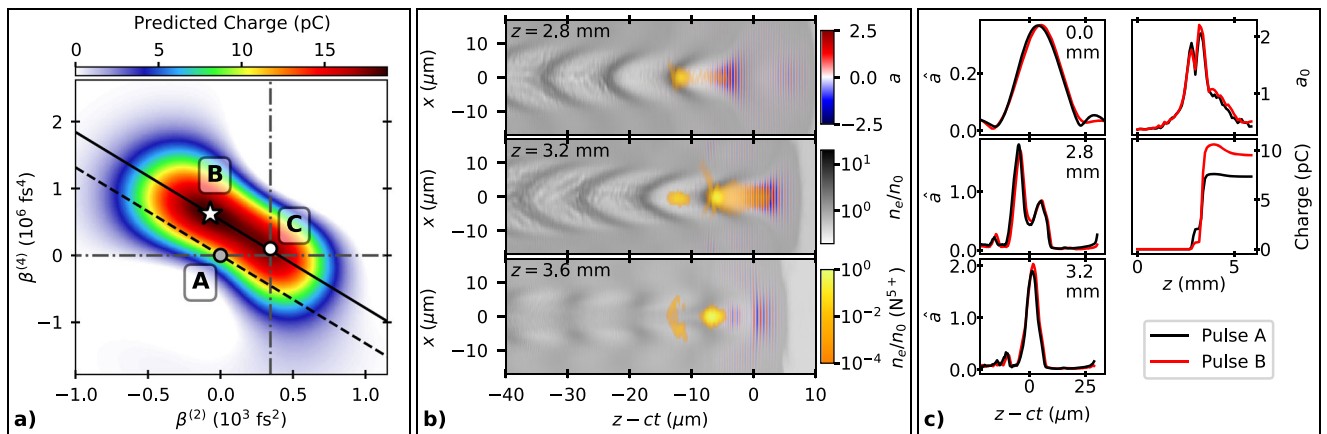

**Fig. 5 Electron beam charge optimization through pulse shaping. a** Surrogate model predicted charge on the $\beta^{(2)} - \beta^{(4)}$ plane at the optimal values of $\beta^{(3)}$ and $f$. Markers show the initial position projected onto this plane, A, and the optimal position, B. Marker C shows the likely end result of sequential 1D optimizations of $\beta^{(2)}$ and $\beta^{(4)}$ when starting from position A. The diagonal lines show the combination of $\beta^{(2)}$ and $\beta^{(4)}$ modifications that maintain an approximately constant pulse shape. **b** Snapshots from a PIC simulation showing the laser normalized vector potential $a$, the electron densities of the background plasma and the electrons released from the two inner ionization levels of nitrogen normalized to $n_0 = 1.2 \times 10^{19}$ cm$^{-3}$. **c** Left: axial laser field envelope at the given $z$ positions and (right) maximum laser field, and total electron beam charge as functions of $z$ position from simulations using the input laser pulse spectral phase coefficients from points A and B from **a**.

ionization of the nitrogen dopant in the plasma, the inner two shells of which require $a \geq 1.9$ to field-ionize. This occurs primarily at $z = 3.2$ mm, where a large fraction of the inner shell electrons are trapped within the accelerating structure.

Figure 5c shows how very small differences in the initial pulse temporal profiles at $z = 0.0$ can grow as a result of the non-linear behaviour. Before entering the plasma, the optimal positively chirped pulse (**B**) had a slightly sharper rising edge ($1/e$ intensity half-width 29 fs) than the unchirped pulse (**A**) (32 fs), but the same peak intensity. Although the FWHM pulse duration only changed by 0.5 fs, this sharper rising edge was caused by significant variations in the spectral phase coefficients. During the initial self-focusing stage ($z = 2.8$ mm), the peak $a_0$ of pulse **A** evolved to be 7% higher than pulse **B**. Subsequently, self-focusing and compression of the leading edge of the laser pulse formed a significantly higher second spike in peak intensity at $z = 3.2$ mm. Here, pulse **A** reached its peak value of $a_0 = 2.1$, whereas pulse **B**, due to its sharper initial rising edge, reached a higher peak value of $a_0 = 2.3$. Due to the threshold behaviour of ionization injection[24], most injection occurred at this point in the laser propagation, with the positively chirped pulse B injecting 40% more charge. Simulations using a pulse with a flat spectral phase but an identical temporal profile to **B** injected the same amount of charge, demonstrating that the key factor was the temporal shape of the pulse intensity profile and not the frequency chirp, in line with previous observations[31].

Figure 5a also shows the consequences if the experiment was instead optimized by two sequential 1D scans of $\beta^{(2)}$ and then $\beta^{(4)}$. In this case, the correlation of these two parameters is not found and the final optimum (**C**) is significantly removed from the true optimum, although only slightly lower in predicted charge; the increase in charge for 1D scans is 87% of the increase between the initial and true optimal positions. The potential loss in performance is compounded with every additional dimension considered, especially where the initial position might be further from the optimum. Comparison of different optimization algorithms within this 4D parameter space (see 'Methods') using Monte-Carlo simulations show that Bayesian optimization significantly outperforms sequential 1D optimizations, as well as optimizations using genetic or Nelder–Mead algorithms. To perform a 4D grid search of this parameter space would take an impractical 14,641 measurements (11 measurements per dimension) to obtain the same performance as the Bayesian optimization algorithm.

## Discussion

In this study, we have presented a Bayesian approach to the optimization and control of LWFAs creating a fully automated plasma accelerator. Through the generation of a surrogate model, the algorithm was able to modify the experimental controls and quickly optimize the generated electron and X-ray beams. Interrogation of one of the generated models also provided physical insight into the dynamics of the electron injection process. It is envisaged that by using this optimization-led approach to learn about complex interactions, unexpected behaviours can be discovered and used to inform the design of better future plasma accelerators. The correct choice of objective function for the optimization algorithm also allows for the nature of the plasma source to be fundamentally altered, enabling a single device to serve many different potential applications. This could be further exploited by using the application itself to provide the objective function, for example coherent x-ray production from a LWFA driven free-electron laser. It is anticipated that the first generation of laser-plasma accelerator user facilities will need to make use of automated global optimization in order to maximize their competitiveness.

## Methods

### Experimental setup

*Plasma source.* The plasma source was a gas cell with initially 200 μm ceramic entrance and steel exit apertures. The rear aperture could translate to vary the cell length in the range of 0–10 mm. The side walls of the cell were glass slides, allowing for transverse probing of the region between the apertures. The cell was filled from below via a tube with an electronically triggered valve, which was opened for 50 ms before the laser arrival time to allow for stabilization of the gas flow. A differential pumping system was used to remove gas after the shot in order to maintain the main vacuum chamber pressure at $\lesssim 10^{-3}$ mbar.

After the plasma source, the depleted laser was removed via a thin tape-based plasma mirror. The electrons and X-rays, generated within the plasma, passed through the thin tape to their respective diagnostics.

*Laser.* The laser was operated with a pulse energy of 245 mJ on target in a pulse duration of ~45 fs. The repetition rate was limited to 1 Hz to avoid the deleterious effects of heat-induced grating deformation[33]. The laser was focused at $f/18$ by a 1 m focal length off-axis parabolic mirror and was linearly polarized in the horizontal plane. The laser had a central wavelength of 803 nm and a FWHM bandwidth of 23 nm.

The on-shot temporal profile of the laser pulse was measured using a small region of the compressed pulse by spectral phase interferometry (SPIDER). The spatial phase of the laser was diagnosed with a wavefront sensor (HASO) using the small (<1%) leakage through one of the beam transport mirrors. A cross-calibrated laser profile monitor was used to measure the total laser energy of each pulse.

*Interferometry.* A ~1 mJ, 800 nm temporally synchronized beam was used as a transverse probe of the gas cell. A 75 mm-diameter 750 mm-focal length collection optic was used, resulting in a minimum resolution of 9.7 μm. A folded wavefront Michelson interferometer was used to provide on-shot measurements of the plasma density when the gas cell length was >1.7 mm.

*Electron spectrometer.* The spectrum of the generated electron beams was measured using a magnetic spectrometer consisting of a permanent dipole magnet with a peak magnetic field of 558 mT, a scintillating Lanex screen ($Gd_2O_2S:Tb$[34]) and an Allied Vision Manta G-235B camera, all sealed in a light-tight lead box. The charge calibration was performed using Fuji BAS-MS2325 image plate. The magnet entrance was 574 mm from the electron source and the total length of the spectrometer was 410 mm. The energy range of the spectrometer, for electrons propagating along the axis was 26–251 MeV.

*X-ray diagnostic.* X-rays were diagnosed with a direct-detection X-ray charge-coupled device (CCD) (Andor iKon-M 934) attached to an on-axis vacuum flange placed 1.23 m from the X-ray source. To prevent laser light from reaching the CCD, two sheets of 12.8 μm thick Mylar foil, coated with 400 nm Al on the front surface and 200 nm Al on the back surface, was used to cover the entrance aperture. This had the additional effect of blocking out 99.6% of X-rays below 1.6 keV (K-edge of Al). The X-ray spectrum was retrieved by comparing the transmission through different materials according to ref. [3]. The materials chosen were (33.5 ± 1.1) μm Al (98%)/Mg(1%)/Si(%1), (29.5 ± 0.3) μm Al(95%)/Mg(5%), (20.15 ± 0.45) μm Mg, (21.85 ± 0.25) μm Mylar and (12.9 ± 0.1) μm Kapton, which were mounted on (12.85 ± 0.25) μm Mylar, and all coated with 200 nm Al to prevent oxidation. Additionally, a 50 μm tungsten filter provided the on-shot background signal.

For the optimal burst of the optimization in Fig. 3, the X-ray spectrum was fitted with a synchrotron spectrum with a critical energy of $E_c = 2.9 \pm 1.0$ keV and contained $(1.9 \pm 0.4) \times 10^4$ photons mrad$^{-2}$ above 1 keV.

**Bayesian optimization algorithm.** The fitting algorithm comprised two independent Gaussian process models. The first took the position $X_m$, mean value $\bar{Y}_m$ and variance $\epsilon_m^2$ of each measurement, and created a model capable of predicting the mean $\bar{Y}(X)$ of the objective function with standard deviation $\sigma(X)$. As the measured values of $\epsilon_m$ are a noisy estimation of the true variance $\epsilon(X)^2$, a second Gaussian process model took the values of $X_m$ and $\epsilon_m$ in order to predict the true standard deviation of the objective function $\epsilon(X)$.

The covariance kernel for both GPR models was expressed as a radial basis function added to a white-noise-function. The physical measurement positions were each individually scaled to values that varied by similar amounts, so that the kernels would be approximately isotropic. The hyperparameters of the kernel were optimized during the fitting process by maximizing the marginal likelihood of each.

The two models were combined in order to provide an estimation of the sampling efficiency at any given position. Per ref. [35], this can be represented by a term:

$$\eta = 1 - \frac{\epsilon}{\sqrt{\sigma^2 + \epsilon^2}} \qquad (1)$$

where $\epsilon$ is the uncertainty of each measurement, whereas $\sigma$ is the SD of the Gaussian process model. This was multiplied by the standard expected improvement acquisition function to produce an augmented acquisition function.

In extensively-sampled regions, as $\sigma$ approaches 0 so does $\eta$. On the other hand, where experimental errors are dominated by the model uncertainty, $\eta$ is close to 1.

In addition, the white-noise kernel adds to the model standard deviation $\sigma$ when it should be counted as part of the experimental error $\epsilon$. This affects the behaviour of the acquisition function, and crucially of the term $\eta$. So when calculating the augmented acquisition function the variance of the white-noise term was subtracted from the model variance and added to the experimental variance.

Finding the maximum of the acquisition function is a further optimization problem, but of a function which has many local optima. Evaluation positions for the acquisition function were selected by a series of line segments generated in random directions through existing samples. Each line segment is uniformly sampled, and the maximum of the acquisition function over all points from all lines was used for the next measurement. This approach was inspired by, but is substantially different from, a published solution to the same problem[36], in which a multi-dimensional optimization problem was reduced to a sequence of 1D optimizations in random directions.

*Computation time for Bayesian optimization algorithm.* Each iteration of the Bayesian optimization algorithm required two computationally expensive steps.

1. Fitting of the Gaussian process models to the measurements.
2. Finding the maximum of the acquisition function.

The execution time of both steps increases approximately linearly with the number of measurements. On a PC with a Intel Xeon Processor E5-1620 v3 3.5 GHz CPU and 64 GB of 2.1 GHz RAM, step 1 took 260 ms and step 2 took 290 ms after 50 measurements. Note that maximizing the acquisition function is another optimization problem which involves multiple evaluations of the acquisition function. For the results of this study, including the execution time given above, 20,000 function evaluations were used to maximize the acquisition function.

**Comparison of optimization algorithms**. Alternative optimization algorithms can be tested using the same surrogate model shown in Fig. 5a, by sampling from the final distribution. These synthetic measurements were then used as the objective function for the alternative optimization algorithm. Each algorithm (except grid search) was performed >100 times and was initialized from a randomized starting point in parameter space. The convergence was calculated by taking the average of the model prediction at each optimal point found by the optimizations as a fraction of the global optimum. The Bayesian optimization model used in the experiment reached an average of 94% of the model optimum with 60 measurements. By comparison, 4× sequential 1D scans achieved 80% convergence using the same number of measurements (15 measurements per axis). A genetic algorithm (SciPy differential evolution[37]) achieved 69% convergence in 60 measurements. A Nelder–Mead algorithm (also from the SciPy library) achieved 34% convergence using 60 measurements. Both the genetic and Nelder–Mead algorithms suffered from the small number of measurements and from the stochasticity of the data; problems which are more easily overcome by the Bayesian approach. A 4D grid search obtained 95% convergence using 14,641 measurements (11 measurements per dimension). It should be acknowledged that the convergence of any of these algorithms could be improved through tuning of the algorithms and their hyperparameters.

**PIC simulations**. Simulations were performed using the PIC code FBPIC. FBPIC uses cylindrical symmetry with azimuthal mode decomposition which is well suited to situations close to cylindrical symmetry. For the simulations in this study, two azimuthal modes were used, over a simulation window of $80 \times 80$ μm in $1600 \times 100$ cells in the $z$ and $r$ directions, respectively. The electron density profile used was based on fluid modelling of the gas density profile using the code OpenFOAM. This gave entrance and exit density ramps that fell to half of the maximum density over a distance of 700 and 850 μm, respectively, and a plateau of uniform density of length 1 mm starting at $z = 2$ mm. The plasma was initialized with $He^{1+}$ and $N^{5+}$ ions with a free-electron species neutralizing the overall charge density. The initial electron density in the plateau was $1.26 \times 10^{19}$ cm$^{-3}$. Each species used $2 \times 2 \times 8$ macro-particles in the $z \times r \times \theta$ directions. Ionization is handled in FBPIC by an algorithm based on ADK ionization rates. The laser pulse was initialized to match the experimental measurements of laser energy, spectral intensity and phase and intensity distribution at the focal plane. The laser pulse spatial phase and intensity distribution were then modified to ensure focusing in vacuum would occur at the start of the density plateau.

## Data availability

The data presented in this paper and other findings of this study are available from the corresponding author upon reasonable request.

## Code availability

The computer code used to perform the augmented Bayesian optimization is available at the online repository zenodo.org with the accession code 4229537.

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

## Acknowledgements

We gratefully acknowledge the hard work of the staff at the Central Laser Facility in the planning and execution of the experiment. R.J.S., J-N.G., M.B., S.P.D.M., Z.N. and M.J.V. S. acknowledge funding from Science and Technology Facilities Council Grant number ST/P002021/1 and the EU Horizon 2020 research and innovation programme grant number 653782. A.G.R.T. acknowledges funding from US NSF grant number 1804463 and US DOE/FES grant number DE-SC0020237. A.G.R.T. and K.K. acknowledge funding from US DOE/High Energy Physics grant number DE-SC0016804. C.T. acknowledges funding from the Engineering and Physical Sciences Research Council grant number EP/S001379/1.

## Author contributions

R.J.S., S.J.D.D., J-N.G., C.I.D.U., A.F.A., C.A., M.B., C.D.B., M.D.B., N.B., J.A.C, P.H., J.K., K.K., S.P.D.M., C.D.M., N.L., J.O., K.P., P.P.R., C.P.R., S.R., M.P.S., A.J.S., D.R.S., A.G.R. T., C.T., Z.N. and M.J.V.S. contributed to the planning and execution of the experiment. R.J.S., J-N.G., C.I.D.U. and M.J.V.S. performed analysis. S.J.D.D. developed the software framework for experimental automation. S.J.D.D., R.J.S. and M.J.V.S. wrote the experimental control and analysis algorithms. S.J.D.D. and M.J.V.S. wrote the Gaussian process regression interface using the Scikit-learn interface. C.A. and M.J.V.S. performed PIC simulations. R.J.S., J-N.G., S.J.D.D. and M.J.V.S. wrote the paper with contributions from C.I.D.U., A.F.A., C.A., M.B., C.D.B., M.D.B., N.B., J.A.C., P.H., J.K., K.K., S.P.D.M., C.D. M., N.L., J.O., K.P., P.P.R., C.P.R., S.R., M.P.S., A.J.S., D.R.S., A.G.R.T., C.T. and Z.N.

## Competing interests

The authors declare no competing interests.
