## [Peer Review File · Nature Communications]

Reviewer #1 (Remarks to the Author):

In my original review, I had two main concerns (1) not an improvement from the state-of-the art result and (2) no new physics. The authors have directly address both points. They argue that their results are "best in class" and that the statistical approach used was required to achieve these results. Although there is a large dataset shown, the majority of the work presented was done with laser powers greater than 5TW and correspondingly show both higher energy and higher charge beams that reported by this manuscript. It is true for the limited data (4-6 points) presented, their results are "best in class." This is marginally acceptable for my interpretation of Nature Communication's standards. On a more positive note, I agree with their argument regarding my original claim of no new physics. They point out that the statistical methods used provided the first fully optimized plasma accelerator and that the quantitative insight into the effects of pulse shape on injection, which could influence future work. I don't think the manuscript has made it significantly over the bar for Nature Communications, but it is well written, the experiments are of high quality, and a new technique has been presented that will clearly be used in future experiments with larger laser systems.

Reviewer #2 (Remarks to the Author):

Dear editor,

The authors proposed a revised version from a previous review stage. I still agree with the various positive comments which have been made during this previous phase concerning the novelty, impact and global clarity of this work. The answers to the referees' questions are complete and interesting, and the modifications provided to the manuscript are satisfactory (although minimalist). Nevertheless, in my view, there are still some (minor) points which could be misleading, and some provided discussions elements which could be beneficial to the main text.

From the answers, it is clearer to me that a direct comparison to the human-based optimisation was not the point of the authors, rather automation from switching on the laser system (I believe this could be stated more clearly in the text). Yet, some expressions let think the contrary (while no clear element in the text support this). In the abstract, "improvements over manual optimisation, enhancing the electron or x-ray yields by factors of 3 or more", this factor for the electrons follows from starting the laser with settings from a previous day (manual) operation, despite of day-to-day laser fluctuations. Hence, this can not conclude to an improvement over manual optimisation. Also page2: "far exceeding that achieved manually" (the corresponding manual achievements have not been reported for comparison).

Concerning the improvement over the state of the art, although it is not claimed by the authors, this is logically expected from a better optimisation procedure on a N-D space. The figure1 of the reply answers well to this concern. In my opinion, this figure should be part of the accessible material. In addition, as the authors stated, this laser system is old, thus there should necessarily be some previous human-based results with this laser, which could be added on this figure. Improvements should then speak for themselves.

Concerning the use of comparisons, I still find uncomfortable with some expressions. Particularly: "a 24 times increase", here the initial condition for comparison is unspecified (and is not relevant. As I suggested, starting from conditions without electrons, which is often the case, would have led to a nonsense infinite increase in energy). Concerning the "five fold increase in x-ray... results in a dramatic increase in the usability", my impression is that the quality of the x-rays is more relevant that just the increased yield (moreover from an unspecific reference, and with "lower pulse energy"). Also the authors do not provide some citation to support their claim that such lasers are usually not adequate for imaging (while they attenuate the x-rays prior the camera).

Concerning the lack of physics raised by reviewer 1, the mention of a sharp-edge and flat spectral phase simulation brings more elements. I would also suggest adding some words on the symmetry axis in figure 5a which advocates for a limited influence of chirp and plasma linear propagation effects. The steeper profile is hardly visible from figure 5c, could the authors provide at least some numerical values to embody the corresponding changes.

Also, some minor points:

Page4, If I am correct, the value of $b(2)$, $b(3)$, $b(4)$ stems from 10 different optimisation runs, yet no uncertainties on these values are provided.

Page5 "The solid and dashed lines ... represents this relationship" (if I am correct, rigorously, only the dashed line corresponds to the formula). Also inexact: "the dashed line shows the case of a fully compressed pulse".

Manuscript title: I wonder if the word "control" is correct since no real time control of drift in laser parameters is addressed.

Reviewer #3 (Remarks to the Author):

This paper, entitled "Automation and control of laser wakefield accelerators using Bayesian optimization", has been revised thoroughly corresponding to the referees' comments. As shown in the figure in the responses to referees, this automation method can provide noticeable benefits to laser-plasma accelerators. The automation of laser-plasma accelerators is a critical issue as the repetition rate of the high power lasers increasing. The automation method can be a very useful method for kHz-repetition-rate LWFAs. Therefore, I would like to recommend the publication of this paper in Nature Communications. Few comments for improving the paper further are as follows.

1) It would be better to mention this automation method improves the charge and energy of laser-plasma accelerators comparing to the current state-of-art results as shown in the figure in the referee response. It can be mentioned with few references done with sub-10 TW laser pulses and gas cells.

2) Defining the axes of figure 1 (b), (c), (d), and (e) is better for easy understanding, even though they are not physical dimensions.

3) The reason for the extremely sensitive dependence of LWFA on the small difference in the laser profile should be clarified further. If there are no effects of laser chirp in the PIC simulations, the stiffness of the laser in the leading edge and skewness of the laser profile can be critical parameters. The sensitive dependence on the laser profile was described by the deformation of the laser pulse during propagation. In this case, the etching speed should be also different for this small change of the laser profile. Can this small profile difference make difference in the etching speed? Also, even though the slightly stiffer rising edge can introduce ionizing injection a bit earlier, but the less stiff case also can inject electrons as the propagation going on. The PIC results explaining the experimental results should be clarified further to support the conclusion.

Point by Point Response To Reviewers Comments

Please find below our point-by-point response to the reviewers comments. Also note that all changes made to the manuscript text since the last version have been highlighted in magenta for ease of identification.

Reviewer #1 (Remarks to the Author):

In my original review, I had two main concerns (1) not an improvement from the state-of-the-art result and (2) no new physics. The authors have directly address both points. They argue that their results are "best in class" and that the statistical approach used was required to achieve these results. Although there is a large dataset shown, the majority of the work presented was done with laser powers greater than 5TW and correspondingly show both higher energy and higher charge beams that reported by this manuscript. It is true for the limited data (4-6 points) presented, their results are "best in class." This is marginally acceptable for my interpretation of Nature Communication's standards. On a more positive note, I agree with their argument regarding my original claim of no new physics. They point out that the statistical methods used provided the first fully optimized plasma accelerator and that the quantitative insight into the effects of pulse shape on injection, which could influence future work. I don't think the manuscript has made it significantly over the bar for Nature Communications, but it is well written, the experiments are of high quality, and a new technique has been presented that will clearly be used in future experiments with larger laser systems.

We thank reviewer 1 for their helpful comments and their second review of the manuscript. We agree that there are not many experiments in the 5 TW range; this reflects the lack of published results with electron beams generated at such low laser powers. However, the experimental results presented in this paper with 5.4 TW are comparable to those from other published works with lasers demonstrating peak powers 2-3 times higher than ours and are above the main trendline shown in the last response.

We thank the reviewer for reconsidering their position relating to the demonstration of new physics within the paper and are glad to hear that they see the potential for this work to influence future experiments.

Reviewer #2 (Remarks to the Author):

Dear editor,

The authors proposed a revised version from a previous review stage. I still agree with the various positive comments which have been made during this previous phase concerning the novelty, impact and global clarity of this work. The answers to the referees' questions are complete and interesting, and the modifications provided to the manuscript are satisfactory (although minimalist). Nevertheless, in my view, there are still some (minor) points which could be misleading, and some provided discussions elements which could be beneficial to the main text.

We thank reviewer 2 for their second evaluation of the manuscript, for the positive points raised and for their recommendations of minor changes which could lead to further improvement in the manuscript. We respond to these points directly below.

From the answers, it is clearer to me that a direct comparison to the human-based optimisation was not the point of the authors, rather automation from switching on the laser system (I believe this could be stated more clearly in the text). Yet, some expressions let think the contrary (while no clear element in the text support this). In the abstract, "improvements over manual optimisation, enhancing the electron or x-ray yields by factors of 3 or more", this factor for the electrons follows from starting the laser with settings from a previous day (manual) operation, despite of day-to-day laser fluctuations. Hence, this can not conclude to an improvement over manual optimisation. Also page2: "far exceeding that achieved manually" (the corresponding manual achievements have not been reported for comparison).

Concerning the improvement over the state of the art, although it is not claimed by the authors, this is logically expected from a better optimisation procedure on a N-D space. The figure1 of the reply answers well to this concern. In my opinion, this figure should be part of the accessible material. In addition, as the authors stated, this laser system is old, thus there should necessarily be some previous human-based results with this laser, which could be added on this figure. Improvements should then speak for themselves.

We entirely agree with the reviewers point that specific numerical comparisons between manual and automatic optimization are difficult to make and will vary on a case by case basis. We point out that the numerical examples given here are not intended to represent such a comparison and are only meant to demonstrate improvements made by the algorithm with respect to its own starting point. However, upon further consideration, we agree that this could be misinterpreted and so to prevent this, we follow the reviewers recommendations to remove these specific comparisons from the text and to qualify other comparisons with published data. Specifically:

- We remove the sentence "*improvements over manual optimisation, enhancing the electron or x-ray yields by factors of 3 or more*" from the abstract

- Pg 1: We also have modified the sentence including *“far exceeding that achieved manually”* to be *“Simultaneous control of up to six laser and plasma parameters enabled independent optimisation of different properties of the source far exceeding that achieved manually with a 5 TW class laser system.”* and we have further included a citation to a published version of the dataset (Mangles:2016lcr) we presented in the response to referees to demonstrate this, as requested by the reviewer.
- Pg 5: the sentence describing the 24 times improvement has been modified to remove the numerical comparison. It now reads *“all five input parameters had to vary significantly to achieve the optimum, an average total beam energy of 0.91± 0.15 mJ”*

Concerning the use of comparisons, I still find uncomfortable with some expressions. Particularly: “a 24 times increase”, here the initial condition for comparison is unspecified (and is not relevant. As I suggested, starting from conditions without electrons, which is often the case, would have led to a nonsense infinite increase in energy). Concerning the “five fold increase in x-ray... results in a dramatic increase in the usability”, my impression is that the quality of the x-rays is more relevant than just the increased yield (moreover from an unspecified reference, and with “lower pulse energy”). Also the authors do not provide some citation to support their claim that such lasers are usually not adequate for imaging (while they attenuate the x-rays prior to the camera).

With regards to the “24 times increase”, we have removed this direct numerical comparison upon the reviewers recommendation (details above).

With regards to the x-rays; we have clarified the statement about the usability of the x-ray source so that it now refers to imaging applications in the multi-keV range, as is required when using plastic and aluminium filters for the laser as in this experiment. We include a reference (Albert PPCF 2016), which contains discussion of LWFA betatron x-ray sources and a table of suitable laser systems.

We have also quantified the lower pulse energy in the main text - “7 % lower average pulse energy”

Concerning the lack of physics raised by reviewer 1, the mention of a sharp-edge and flat spectral phase simulation brings more elements. I would also suggest adding some words on the symmetry axis in figure 5a which advocates for a limited influence of chirp and plasma linear propagation effects. The steeper profile is hardly visible from figure 5c, could the authors provide at least some numerical values to embody the corresponding changes.

We have included more discussion around fig 5a and the physics observed in the simulations (concurrent with the comments raised here and comments raised by reviewer 3 below). Numerical values have also been added to quantify the profile steepness,

Also, some minor points:

Page4, If I am correct, the value of $b(2)$, $b(3)$, $b(4)$ stems from 10 different optimisation runs, yet no uncertainties on these values are provided.

The values of $b(2)$, $b(3)$, $b(4)$, etc were actually determined from the model created by combining the 10 runs. As such they do not have an error associated with them. We have therefore moved the sentence giving these values to the relevant part of the manuscript, the 'Exploring the models' section, where the optimal parameters are discussed. For the section on convergence of the optimised charge it is more appropriate to discuss the mean and standard deviation of the electron beam charge and so we have included this instead.

Page5 "The solid and dashed lines ... represents this relationship" (if I am correct, rigorously, only the dashed line corresponds to the formula). Also inexact: "the dashed line shows the case of a fully compressed pulse".

Both lines correspond to the approximate relationship given relating changes in $b(4)$ to $b(2)$ given in the previous sentence, the lines being separated by a horizontal shift.

With regards to the phrase "the dashed line shows the case of a fully compressed pulse", the reviewer is correct that it is inexact. The line represents combinations of $b(2)$ and $b(4)$ which cancel out the group delay at $\pm \Delta \omega$. We have modified the text and included a formula to clarify this point and the meaning of each line.

Manuscript title: I wonder if the word "control" is correct since no real time control of drift in laser parameters is addressed.

The word 'control' in the title refers to the control of the input parameters in order to change the nature of the laser-plasma source and its output beam properties. As this is an important aspect of the work, we would prefer to keep it in the title.

Reviewer #3 (Remarks to the Author):

This paper, entitled “Automation and control of laser wakefield accelerators using Bayesian optimization”, has been revised thoroughly corresponding to the referees’ comments. As shown in the figure in the responses to referees, this automation method can provide noticeable benefits to laser-plasma accelerators. The automation of laser-plasma accelerators is a critical issue as the repetition rate of the high power lasers increasing. The automation method can be a very useful method for kHz-repetition-rate LWFAs. Therefore, I would like to recommend the publication of this paper in Nature Communications. Few comments for improving the paper further are as follows.

We would like to thank reviewer 3 for their second evaluation of the manuscript and for their recommendation for publication in Nature Communications. We agree that optimisation is a critical issue and that techniques such as those presented in this paper will be built upon and will find use in future high-repetition-rate plasma accelerators.

1) It would be better to mention this automation method improves the charge and energy of laser-plasma accelerators comparing to the current state-of-art results as shown in the figure in the referee response. It can be mentioned with few references done with sub-10 TW laser pulses and gas cells.

We agree with reviewer 3 (and also reviewer 2) that the data presented in the response to reviewers should be made available in the paper. Thus, we have now cited the original paper which collated the laser wakefield data for the plots presented in the response to reviewers. This includes several references related to LWFA with sub-10 TW laser pulses and to experiments with higher laser powers, to which our electron energy and charge are comparable.

2) Defining the axes of figure 1 (b), (c), (d), and (e) is better for easy understanding, even though they are not physical dimensions.

In this case we disagree that axis labels would add any extra understanding. Because there are several different variables plotted (detailed in the caption) the vertical axis would simply be labelled ‘Y’. Additionally, as the horizontal axis represents a general 1D space it would be labelled ‘X’. Rather than introducing extra unused variables such as these, we would like to leave the axes clear. We have endeavoured to ensure that all of the relevant information is in the caption.

3) The reason for the extremely sensitive dependence of LWFA on the small difference in the laser profile should be clarified further. If there are no effects of laser chirp in the PIC simulations, the stiffness of the laser in the leading edge and skewness of the laser profile can be critical parameters. The sensitive dependence on the laser profile was described by the deformation of the laser pulse during propagation. In this case, the etching speed

should be also different for this small change of the laser profile. Can this small profile difference make difference in the etching speed? Also, even though the slightly stiffer rising edge can introduce ionizing injection a bit earlier, but the less stiff case also can inject electrons as the propagation going on. The PIC results explaining the experimental results should be clarified further to support the conclusion.

The pulse evolution in our case is relatively complicated, due to the high plasma density relative to the pulse length. Due to this the pulse simultaneously undergoes self-modulation, self-compression, pump-depletion and self-focusing effects. These are all linked together by the plasma response and so they are all varying spatially and temporally throughout the interaction. The formation of a single high intensity spike late in the interaction was observed to be responsible for the majority of the injected charge, and forms as a result of self-modulation and self-focussing effects. We have added some additional discussion and citations in the paper to try and make this description clearer and we thank the reviewer for the interesting questions.

Other Changes to the Manuscript to Comply with Nature Communication Style Guidelines

We have also made the following changes to ensure compliance with Nature Communications style guidelines as detailed in

<https://www.nature.com/documents/ncomms-formatting-instructions.pdf>

- We have introduced the correct section headings
- We have changed current section headings to subheadings
- We have reordered the methods and references to comply with the Nature Communications section ordering.

Reviewer #2 (Remarks to the Author):

The authors have addressed my concerns, improved the physical discussions and removed misleading points. Therefore, I would like to recommend this significant piece of work to Nature Communications.

Reviewer #3 (Remarks to the Author):

This paper has been revised corresponding to the referees' comments for the publication in Nature Communications. The automation of LWFA is an interesting and important subject, even though the quality enhancement of the electron beam could be considered as marginal for Nature Communications. This paper presented clearly the machine learning algorithm can optimize multiple control parameters to enhance an aspect of electron beam quality from LWFA. I think this work is deserved to be published in Nature Communications, but the physics behind extremely sensitive dependence on the laser profile is not fully clarified yet. I agree that laser propagation in plasma is a complicated phenomenon, but there should be detailed explanations because this tiny difference in laser profile is very difficult to control. Since the main topic of this paper is the automation of LWFA with a machine learning algorithm, I would suggest detailed investigations on this sensitive dependence of LWFA on laser profile further to be published later elsewhere. Consequently, I continue to recommend this paper to be published in Nature Communications.

Point by Point Response To Reviewers Comments

Reviewer #2 (Remarks to the Author):

The authors have addressed my concerns, improved the physical discussions and removed misleading points. Therefore, I would like to recommend this significant piece of work to Nature Communications.

Reviewer #3 (Remarks to the Author):

This paper has been revised corresponding to the referees' comments for the publication in Nature Communications. The automation of LWFA is an interesting and important subject, even though the quality enhancement of the electron beam could be considered as marginal for Nature Communications. This paper presented clearly the machine learning algorithm can optimize multiple control parameters to enhance an aspect of electron beam quality from LWFA. I think this work is deserved to be published in Nature Communications, but the physics behind extremely sensitive dependence on the laser profile is not fully clarified yet. I agree that laser propagation in plasma is a complicated phenomenon, but there should be detailed explanations because this tiny difference in laser profile is very difficult to control. Since the main topic of this paper is the automation of LWFA with a machine learning algorithm, I would suggest detailed investigations on this sensitive dependence of LWFA on laser profile further to be published later elsewhere. Consequently, I continue to recommend this paper to be published in Nature Communications.

We would like to thank both reviewers for their review of the revised manuscript and for their positive and helpful comments.